# The Role of Cyclodextrin in the Construction of Nanoplatforms: From Structure, Function and Application Perspectives

**DOI:** 10.3390/pharmaceutics15051536

**Published:** 2023-05-19

**Authors:** Chengyuan Xing, Xiaoming Zheng, Tian Deng, Ling Zeng, Xin Liu, Xinjin Chi

**Affiliations:** 1Institute of Sports Medicine and Health, Chengdu Sport University, Chengdu 610041, China; 2The Seventh Affiliated Hospital, Sun Yat-sen University, Shenzhen 518107, China; 3Department of Laboratory Medicine, The Second Xiangya Hospital of Central South University, Changsha 410008, China

**Keywords:** cyclodextrin, host-guest inclusion, drug delivery, interactions, application

## Abstract

Cyclodextrins (CyDs) in nano drug delivery systems have received much attention in pursuit of good compatibility, negligible toxicity, and improved pharmacokinetics of drugs. Their unique internal cavity has widened the application of CyDs in drug delivery based on its advantages. Besides this, the polyhydroxy structure has further extended the functions of CyDs by inter- and intramolecular interactions and chemical modification. Furthermore, the versatile functions of the complex contribute to alteration of the physicochemical characteristics of the drugs, significant therapeutic promise, a stimulus-responsive switch, a self-assembly capability, and fiber formation. This review attempts to list recent interesting strategies regarding CyDs and discusses their roles in nanoplatforms, and may act as a guideline for developing novel nanoplatforms. Future perspectives on the construction of CyD-based nanoplatforms are also discussed at the end of this review, which may provide possible direction for the construction of more rational and cost-effective delivery vehicles.

## 1. Introduction

Although traditional drugs show good therapeutic promise, they also face a series of drawbacks such as poor water-solubility, instability, short circulation time, unspecific targeting, and low biocompatibility, and thus they struggle to be as effective as expected in clinics [1,2]. Hydrophobic drug molecules can accumulate in fat tissues, impairing patient recovery [3]. Additionally, 70% of new drug candidates are hydrophobic, which requires great efforts to formulate these water-insoluble drugs [4]. Drug administration in conventional dosage forms can result in unwanted therapeutic outcomes, such as drug toxicity/drug inefficiency and uncontrolled drug delivery. High drug loading and sustained/controlled release into targeted sites have led to the development of advanced drug delivery systems. Various strategies are used to improve pharmaceutical or biopharmaceutical delivery systems, such as the interaction principle, assembly technologies, and/or targeted strategies [5]. Researchers have also developed many kinds of nanoplatforms (such as polymer, lipid, amino acid, polypeptide, and inorganic-based platforms) for drug delivery, and dramatic advances have been made in chemistry, materials science, clinical practice and biotechnology [6,7]. Many of the drug delivery systems have focused on controlled release therapeutics at appropriate times. They can be endowed to target specific locations within the body by active or passive targeting, thereby reducing the number of drugs used to achieve equal therapeutic effect, along with alleviating side effects to the patients [8,9,10]. The drug delivery systems also allow for more specific drug targeting and administration by tailoring the physicochemical and pharmacokinetic/pharmacodynamics properties of the original drugs to maximize therapeutic benefits. 

In the field of drug delivery, the following improvements should be sought: (i) improvement in treatment effect; (ii) attaining a minimum in drug leakage/toxicity in normal tissues or organs; and (iii) reduction in the intake dosage [11]. Several improvements can be achieved in a structural manner derived from the CyDs-based drug delivery system. In 2000–2004, FDA introduce α-CyD, β-CyD, and γ-CyD into the generally regarded as safe (GRAS) list for use as a food additive. CyDs are highly safe for animal and human administration. The first patent for CyDs in pharmaceutical formulations dates back to 1953 [12]. Over 70 CyDs-related programs are being studied for clinicians. At least six types of CyDs and over 89 pharmaceutical products based on CyD complexes have been for clinics [13,14]. Oral, nasal, pulmonary, ocular, and parenteral routes have been used to deliver CyDs. α-CyD uptake for 13 weeks at 13.9 and 12.6 g per kg body weight per day did not cause any toxicity or adverse health effects [15]. Chemically modified CyDs have been synthesized to improve native CyDs’ solubility, inclusion capability, controlled drug delivery capacity, and toxicity. The potential of CyDs within drug delivery systems have motivated us to conduct in-depth exploration. However, there are still confusion about how to neatly utilize CyDs in the drug delivery system and how to use CyDs to achieve incredible therapeutic effects. 

In this review, interesting directions and their potentials in nanoplatforms based on CyDs have been discussed. We will also demonstrate engineered drug delivery systems designed through representative supramolecular interactions, covalent interactions, and other interesting strategies related to CyD functions. Meaningful self-assembly, gene delivery, and stimulus responsive strategies have been listed and discussed. The current challenges in the field of drug delivery, breakthroughs in CyDs research, as well as future considerations and opportunities for the translation of CyDs-based materials into clinical practice, are also all highlighted. 

## 2. Structure

There are three different kinds of hydroxyl groups in CyDs, namely 2-OH, 3-OH and 6-OH (Figure 1). Generally, these hydroxyl groups of CyDs provide active sites for chemical modification. 6-OH has better availability and nucleophilicity than 2-OH and 3-OH, so it is possible to react with a wide range of active groups to modify the molecules. Since the secondary hydroxyl groups (2-OH and 3-OH) are more acidic, they are usually inactive [16]. Highly reactive reagents such as the isocyanate group could non-selectively react with all hydroxyl groups, whereas less reactive reagents are selective in their reaction with 6-OH [17]. Natural CyDs have been further derivatized by organic synthesis in recent years, which has greatly improved their bonding ability, solubility, and stability of the inclusion complex, thus expanding its application in molecular recognition and molecular assembly.

Unlike linear counterparts (e.g., starch), CyDs are resistant to enzymatic degradation by β-amylases, but gradually hydrolyze by α-amylases, particularly γ-CyD, which is susceptible to rapid degradation by salivary and pancreatic enzymes. The affinity of α-amylases for CyD substrates increased in the order of γ-CyD > β-CyD > α-CyD. α-Amylases cleave internal α-D-(1, 6) and α-D-(1, 4) of glycosidic bonds and show a tendency towards CyDs [18]. CyDs can meet many desired characteristics of pharmaceutical excipients and are not limited to their inert and harmless structure. In addition to being cost-effective, pharmacologically inert, and stable, CyDs do not interact with drug molecules generally [19].

## 3. Function

### 3.1. Noncovalent Interactions

There are many noncovalent interactions between CyD and other molecules, such as van der Waals interactions, hydrophobic interactions, π-π interactions, electrostatic interactions, host-guest interactions, metallic coordination, and hydrogen bonds. CyD absorption frequency is reduced with other molecules due to the noncovalent interactions, which reduce the force constants of the corresponding bonds and lower the energy of molecules. Although there are a series of interactions and even though there are paradoxical effects between CyD and other molecules, certain dominant interactions lead to complexation [20]. In a study by Jian et al., it was shown that γ-CyD can be absorbed by nanotubes/ropes via van der Waals forces and weak electrostatic interactions using optical absorption spectrum measurements [21]. Pawar et al. have reported that the suitable size of artemether and the interaction with hydroxypropyl-β-CyD (HP-β-CyD) led to its efficient entrapment and increased inclusion constant [22]. Meanwhile, the methyl group of HP-β-CyD also helps to drive artemether towards the lipophilic cavity because of hydrophobic interaction.

CyDs are quite compatible with water molecules because there are three kinds of -OH groups that can form hydrogen bonds. The water molecules in the cavity bind as complex water while bind as crystal lattice water outside. In an aqueous state, there is a balance between free molecules and CyD molecules. The solubility of natural CyDs is also influenced by the strong binding of CyD in the crystal form. By substituting -OH groups with hydrophobic groups of CyD, intramolecular hydrogen bonding weakens, leading to amorphous solids.

Generally, several parameters like size, polarity, and hydrophobicity are required in the formation of stable host-guest inclusion complexes in aqueous solutions. Due to the absence of covalent bonds, unstable bound water can be released from the inner cavity. The reason for this is due to the dynamic replacement of a suitable guest molecule into the cavity and ultimately stabilizing the entropy. As incoming guest molecules bind, bound water is completely or partially replaced by incoming guest molecules. Organic guest molecules enter the CyD cavity through the complexation interactions in an aqueous solution due to their shape, size, and polarity. To determine the stability of the bond, several parameters should be considered, such as size/shape matching, hydrophilicity/hydrophobicity, and electrostatic interaction. Moreover, one of the key essentials is that the formation of the host-guest inclusion should be in the presence of water molecules [23]. So far, scientists have synthesized a large number of CyD derivatives to improve the properties of unsatisfying drug molecules. Based on noncovalent interactions, CyDs serve a variety of purposes, including removing lipids, increasing solubility, and reducing side effects. Please refer to the following re-searchers for more information on the non-covalent interactions between CyDs: Szejtli et al. and Lorenzo et al. [24,25,26].

### 3.2. Functional Modification of the Active Hydroxyl Groups

CyDs have been widely used in pharmaceutical fields, mainly due to their negligible toxicity, ideal biocompatibility, and non-immunogenicity [27,28,29]. Although bare CyDs have a variety of roles in drug delivery, the incorporation of pre/post modification has widened the application range and helped to build many kinds of novel materials. For example, due to the polyhydroxy structure, CyDs can react with a wide variety of reagents to obtain water-soluble CyD derivatives. Different chemical modifications have been proposed by grafting reactive groups at different positions (primary 6-OH, 2, 3-OH, or both -OH). The as-formed supramolecular aggregates are suitable for drug delivery applications. Through functional modifications, CyDs can introduce ionic groups, hydrophobic groups, and other active linkers (such as azide groups, *p*-toluene sulfonyl groups, halogen groups, sulfhydryl groups, etc.). The modification of ionic groups could act as gene delivery vectors or permeation-enhancing excipients. CyDs could also be modified to connect chemotherapy drugs, receptors, and targeted agents. The intramolecular hydrogen bonding and moderately high crystal lattice energy of β-CyD result in low water solubility. The aqueous solubility of the derivative is dramatically improved when any of the hydrogen bond-forming hydroxyl groups are replaced.

## 4. Application

Based on their unique structure, CyDs have shown different roles in drug delivery. CyDs are regarded to be an efficient and accessible functional unit for the construction of biomedical engineering materials accounting for their biosafety and modifiability. On the one hand, because of the high adaptability to guest molecules, the inner cavity of CyDs could accommodate a variety of different pharmaceutical molecules, such as stimulus-responsive units and biomolecules. On the other hand, its functional groups could interact with surrounding molecules through various intermolecular interactions. Based on covalent and non-covalent interactions, CyDs have been widely explored by researchers in the prospect of physicochemical characteristics alteration of the drugs, therapeutic promise, stimulus-responsive switch, self-assembly capability, and fiber formation.

### 4.1. Physicochemical Characteristics Alteration of the Drugs

The included guest molecules ranged from small molecules, ions, and proteins to polymers with certain shape and size requirements [30]. Consequently, variable CyDs complexes have played an important role(s) in drug delivery systems, e.g., increase solubility, enhance the transdermal effect, realize controlled release, construct the self-assembly structure, and introduce a stimulus-responsive block [31,32,33]. Modern drug delivery systems use CyDs as ideal supramolecular hosts for the preparation of carriers in the form of micelles, vesicles, hydrogels, and metal-organic frameworks [34,35,36,37].

#### 4.1.1. Without Any Modification

CyDs (i.e., α-CyD, β-CyD, γ-CyD, and hydroxypropyl-β-CyD) without any further modification are still attractive for pharmaceutical scientists for their industrial and clinical potential. To achieve a spatiotemporal control of release, CyD is being introduced into the drug delivery system to modify the pharmacokinetic profile. The interactions between CyDs and guest molecules (i.e., hydrogen bond, π-π interactions, ionic and dipolar interactions, hydrophobic force, host-guest, and van der Waals interactions) contribute to the formation of the inclusion behavior [30]. Xu et al. prepared the intercalation of anion carboxymethyl β-CyD (CM-β-CyD) into layered double hydroxide (LDH) using an anionic exchange method [38]. Meanwhile, the inner cavity of β-CyD could also retain drugs like dexamethasone (DEX) and avoid leakage. For eye drops, this drug delivery system penetrated the ocular surface to the posterior segment. Compared with DEX-CM-β-CyD single carrier drug delivery system, DEX-CM-β-CyD@LDH has prolonged anterior corneal residence time, enhanced cell permeability, and increased bioavailability in the posterior segment of the eye. The CM-β-CyD molecule acted as a host molecule with the guest molecule DEX, and it was also included in LDH by ionic interaction. 

#### 4.1.2. Pro or Post Modification of CyDs

With the in-depth investigation of drug delivery demands, bare CyDs without any further modification do not seem sufficient to meet the versatile needs. Thus, material engineers have put forward pre- or post-modification proposals to endow extra roles for CyDs [39,40]. Liu et al. have designed antimitotic peptide-decorated permethyl-β-CyD for the inclusion of porphyrins [41]. The introduction of permethyl-β-CyD improves water solubility of the loaded porphyrins and helps to promote cell apoptosis. The reactive hydroxyl group in permethyl-β-CyD corporates with antimitotic peptide also forms a hydrophilic part relative to permethyl-β-CyD/porphyrin-inclusion complex (IC). Pawar et al. have loaded two different drugs based on β-CyD and HP-β-CyD into nanosponges [22]; the two loaded drugs, artemether and lumefantrine, were both included in the cavity of β-CyD and their solubility and stability have been improved. From another prospect, Kang et al. introduced two kinds of β-CyD (cysteinyl-β-CyD and ethylenediamine-β-CyD, EDA-β-CyD for short) and immobilized on the surface of SiO_2_@Ag NPs [42]. Both DOX-loaded cysteinyl-β-CyD and EDA-β-CyD showed significant cell toxicity to cancer cells, but they are different in drug loading quantities and release behavior. However, the detailed release kinetics of the two β-CyDs should be further investigated. In addition, this work also showed potential for controlled release of drugs based on the two kinds of CyDs. Another interesting work presented by Yunus Basha et al. has demonstrated that two tuberculosis drugs, namely rifampicin and levofloxacin, which show different bacteria-killing behavior, were complexed with β-CyD in one system [43]. By conjugating β-CyD onto hexachlorocyclotriphosphazene (HCTP) via click chemistry reaction, the resultant CyD was synthesized. The multivalent hydrophobic, pH-responsive CyD host material and the multivalent hydrophilic guest macromolecule formed the NPs. The pH-responsive host-guest NPs can be triggered by proton and transformed into hydrogel. A hydrogel barrier formed in situ protects mice against gastric injury caused by ethanol or drugs (Figure 2I,II). Moreover, this type of NPs can be used as triggerable, transformable, and sustainable nanovehicles for therapeutic agents, thereby typical inflammations can be effectively treated [44].

Another meaningful example comes from the inclusion complex of mannose-modified γ-CyD with regorafenib (Figure 2II). As a hydrophobic drug, regorafenib can be included in the cavity of γ-CyD. Meanwhile, the chemically modified mannose could also target the mannose receptor and promote cellular internalization. Together with its self-assembly capability, the system achieves ideal drug delivery with the least cost [45].

The drug delivery behavior was improved by introducing two functional groups. Sun et al. modified the hydroxyl groups of HP-β-CyD with carboxyl groups of biotin and used arginine as the functional spacer [46]. Biotin could interact with the biotin receptor and enhance the endocytosis of the NPs [47]. The introduction of biotin and arginine could increase cellular uptake and result in enhanced anticancer activity. 

Apart from this, scientists also try to combine CyDs with reactive linkers and integrate them into polymer by grafting or crosslinking method. Diao et al. prepared β-CyD polymer and inclusion with curcumin [48]. The result shows that β-CyD polymer improves water dispersity of curcumin in water. The curcumin-β-CyD polymer in aqueous solutions was expected to enhance the hydrophilicity and bioavailability of curcumin. 

Another fascinating point is the chemical modification of inorganic or carbon-based materials [49]. The abundant -COOH, -OH, and -C=C- make it possible to introduce different compounds, including CyDs. As an example, graphene oxide/polymer brush nanocomposites were prepared to deliver hydrophilic (doxorubicin, DOX) and hydrophobic (methotrexate, MTX) drugs together. Polymerized acrylated β-CyD was used as a main monomer which plays a role in improving dispersibility and forming inclusion complexes with hydrophobic MTX [50]. Carbon-based nanomaterials like single-walled carbon nanotubes (SWCNTs) faced a series of limitations of poor biocompatibility and high biological toxicity. This is because of their high hydrophobic surface, low functionality, and large particle size connected by van der Waals force and strong π-π interactions between the separate parts. Now that the physicochemical properties can be improved by host-guest inclusion of CyDs, chemical modification of CyDs could also eliminate or weaken those limitations. Liu et al. grafted β-CyD onto the surface of SWCNTs, which improved the biocompatibility of SWCNTs and also helped to load hydrophobic drug [51]. Besides π-π interactions influencing the release behavior, host-guest interactions contributed to the pseudo-second-order release of drugs, which may provide a benefit to anticancer treatment. 

A drawback of CyDs about inclusion drugs is the release of drugs from their cavities after in vivo administration. CyD inclusion complexes rapidly release drugs after oral and parenteral administration. Dilution drives the release of weakly and moderately bound drugs after parenteral administration, whereas for the firmly bound drugs (with a binding constant of 10^4^ M^−1^), the major factor driving the dissociation is the binding of drugs to plasma and tissues [52]. The formation of NPs can be influenced by various interactions and cannot dilute easily, so this realizes sustained release [53].

### 4.2. Therapeutic Promise

Among the reservoir of the guests, biomolecules derived from organisms could also be included in the cavity in theory. For instance, cholesterol is a good example of a typical biological molecule that is involved not only in the formation of the membranes of cells and organelles but also in the synthesis of bile acids and vitamin D. Cholesterol is the major lipid ingredient of the plasma membrane and is usually ubiquitous in most other organelles. Cholesterol plays important roles in maintaining its fluidity and permeability in the plasma membrane. The CyDs with high affinity for lipophilic compounds can act as lipid carriers instead of endogenous transporters in disease states when lipophilic compounds accumulate abnormally in the body. It has been reported that endosomal and lysosomal membranes (organelle membranes) damage could induce autophagy; CyDs could interact with organelle membranes and extract cholesterol out of the membranes, which might trigger autophagy [54]. 

To verify this hypothesis, Yamada et al. constructed a liposome-type nanodevice for methylated-β-CyD polyrotaxane (PRX) delivery. The modifications of the functional cationic group such as octaarginine (R8, RRRRRRRR) or the S2 peptide (S2, Dmt-D-Arg-F-K-Dmt-D-Arg-F-K), are beneficial to cellular uptake and mitochondrial targeting activity. As a result, the S2 targeted nanomaterial was internalized efficiently by cells, reaching mitochondria followed by autophagy, even with serum in the medium. The inclusion of cholesterol was also approved by Peter et al., who used HP-β-CyD to treat atherosclerosis by dissolving cholesterol crystals [55]. Furthermore, PRX-based nanomaterial was effective in diminishing the cholesterol pool within the liver, spleen, and kidney at 10- to 100-fold doses lower than monomeric HP-β-CyD. PRX scaffolds with different physiochemical properties contributed to structure-activity relationships difference wherein the number of CyD and the type of axle polymer appear to be a large impact on the resultant therapeutic effect [56]. 

Similar to the formation of host-guest inclusion with cholesterol, another example was that HP-β-CyD forms host–guest inclusion with N-retinylidene-N-retinylethanolamine (A2E, a molecular related to macular degeneration). A study has shown that supramolecular therapeutics are powerful candidates to treat macular disease by removing toxic metabolites from host-guest inclusions [57]. Matsuo et al. also demonstrated that HP-β-CyD has been useful for clinical Niemann-Pick disease type C treatment [58].

There are other cases presented by Jana et al., which illustrated CyDs could bind with intracellular tubulin/microtubule [59]. CyDs, especially α-CyD, inhibit tubulin polymerization rate in vitro and interact; with inner microtubules; depolymerization of the microtubules by α-CyD produced intracellular soluble tubulin through the lysosomal pathway. Detailed interaction sites are Asp179, Val177, Tyr210, and Asn329 amino acid partners of the tubulin with -OH group of α-CyD forming hydrogen bonding between α-CyD and the tubulin. Besides this, α-CyD could also include the hydrophobic drugs into the cavity of CyDs to depolymerize tubulin/microtubule further.

### 4.3. Stimulus Responsive Switch

Stimuli-responsive materials can react to surrounding variations and reply properly, and thus it is possible to mimic the responsive behavior from natural and physiological stress [60]. Chemical-responsiveness can be achieved easily through reversible interactions between CyDs and guest molecules. CyDs can also be functionalized by covalent conjugation or supramolecular recognition to introduce additional responsive moieties for chemical or physical stimuli-responsive [61]. Such a possibility enables us to fabricate a stimulus-responsive nanoplatform. Moreover, an ideal stimulus that works in biological systems requires several necessaries, such as good biocompatibility, appropriate responsiveness, and a positive pathway to treatment [62,63]. 

The development of many cytotoxic drugs has led to the success of drug delivery systems, which further improves the treatment results and the quality of life of patients. However, in some conditions, the lack of selectivity especially for cancer treatments remains an important problem, resulting in potentially life-threatening systemic side effects [64]. An ideal drug delivery system can accumulate the desired drug concentration at the targeted position with decreased systemic exposure or minimal unwanted enrichment, thus avoiding side effects. Based on abnormal physiological conditions from the pathological area, such as pH, reactive oxygen species (ROS), glutathione (GSH), enzymes, and biomarkers, together with external light, thermal and magnetic field, stimulus-responsive drug carriers are developed to trigger the drug release in response to variation of environmental factors [65,66,67,68,69]. 

It is suggested that stimulus-responsive switches can be used for drug and gene delivery, as the switch becomes an active part of the therapeutics instead of just a carrier. The many classes of the molecules are utilized for the fabrication of stimuli-responsive switch, and active or passive drug targeting [2]. CyDs could form host-guest inclusion with a variety of stimulus-responsive molecules [70]. Thanks to their good biocompatibility, low toxicity, and self-assembly properties, CyDs-based stimulus-responsive switches have been regarded as potential building blocks and have been well-researched. Furthermore, host-guest inclusion can enable non-covalent interactions between two groups, simplifying the synthesis process and reducing workload. Ferrocene (Fc), phenylboronic acid pinacol ester, azobenene (Azo) and benzimidazole are four representative stimulus responsive guest molecules. Table 1 is CyDs-based host-guest inclusion stimulus responsive studies in recent 5 years. 

The high spatial and temporal resolution of light makes it a unique stimulus for dynamic self-assembly. One of the most frequently used photoswitchable guest units for CyDs are Azo groups (Figure 3I). Azo and its derivatives demonstrate attractive properties to realize reversible isomerization of trans and *cis*-isomers upon external photoirradiation of ultraviolet, visible, or thermal light, and have been widely studied as photoactive molecules. In the process of changing from trans to *cis*, carbon distances on both sides of double bonds decrease from 0.9 nm to 0.55 nm, and dipole moments increase from 0 to 3 D. The two isomers show different inclusion behavior when faced with α-CyD or β-CyD. In particular, trans-Azo can be strongly bound by α-CyD and form a 1:1 inclusion complex with a binding constant of 2.0 × 10^3^ M^−1^, but *cis*-Azo can be rapidly released from the cavity since its binding constant 3.5 × 10 M^−1^ [70]. Light exposure varied the inclusion behavior between Azo and CyD, and this makes it able to be an ideal candidate for photo-controlled drug release.

Proper ROS is essential to keep the life activities of organisms, whereas the overexpression of ROS is related to various diseases, including cancer, inflammation, heart failure, and neurodegenerative diseases. The difference of ROS levels between pathological and normal tissues and even the intracellular and extracellular environment makes it possible to develop redox-responsive nanoplatforms. Phenylboronic acid pinacol ester is a typical ROS responsive molecule (Figure 3II). Particularly, its C-B bond can be cleaved in a high ROS atmosphere, and after this electron transfer contributes to quinone rearrangement to produce the phenol and boronic pinacol ester. Once included in the cavity of β-CyD, the host–guest inclusion complex is relatively stable, and when stimulated by ROS, the C-B bonds can be cleaved. Following this, the as-formed relatively stable structure is disrupted [77]. Phenylboronic acid pinacol ester, also characterized by its high glycosensitivity, has attracted widespread attention due to its capability for forming reversible ester bonds through competitive reactions with *cis*-diols in many saccharides. Xu et al. designed a smart drug carrier that can load insulin and responsive to the variation in blood glucose levels [83]. A reversible phenylboronic acid group-modified CyD (β-CyD-EPDME) insulin carrier was prepared by combining two popular molecules through a simple synthetic procedure. The detached phenylboronic acid moiety triggered by glucose can enter into the β-CyD cavity and form a host–guest complex, and the encapsulated insulin in the cavity can be driven out. Combined with exogenous glucose oxidase, it will produce a large amount of H_2_O_2_. Thus, this safe and glucose-derived H_2_O_2_ responsive drug carrier shows the potential for use in the treatment of diabetes.

By the introduction of Fc motifs, CyD-based carrier becomes sensitive to various redox agents, e.g., Fe^3+^, H_2_O_2_, and sodium hypochlorite (NaClO); this can also be achieved by electrochemical oxidative method. Concerning interactions with Fc derivatives, β-CyD showed the highest binding stability, with a formation constant of 2.2 × 10^3^ M^−1^ [89]. Meanwhile, Fc^+^ cannot be included in the cavity of CyDs [78]. Therefore, by giving redox stimulus to Fc, the as-formed self-assemblies based on CyD-Fc interactions could also become unstable and release the loaded drugs.

The reversible conversion from Fc to Fc^+^ experiences a redox process. Due to the high GSH and ROS levels, such conversion is typically investigated in cancer therapy. In organisms, reduced GSH and oxidized H_2_O_2_ in an acid atmosphere could help to realize the conversion [90,91]. One of the important products in this conversion is toxic ·OH. On one hand, increased ROS could increase the oxidative stress of cancer cells; On the other hand, toxic ROS could kill cancer cells directly. This reversible conversion property is also named the Fenton reaction and could in theory remove cancer cells effectively. Therefore, Fc could play a therapeutic role in cancer treatment.

Similar to Fc, benzimidazoles can also be used as theranostics for clinical treatment. A series of efficacies of benzimidazoles have been verified, such as anti-inflammatory, antalgic, antimicrobial, antiviral, anthelmintic, antiproliferative, anti-hypertensive, and anti-infective activities [92,93,94,95]. Its derivatives also inhibit chemokine receptor, interleukin 2-inducible T cell kinase, and lymphocyte tyrosine kinase, and many scientists have exploited its anticancer potential [96]. 

Abnormal pH in the pathological region promotes the development of pH responsive drug delivery systems. By introducing alkyl guests with protonated nitrogen atoms into the CyD skeleton, pH-sensitive drug delivery systems may be achieved. At neutral pH, benzimidazole has a binding constant of 1.6 × 10^3^ M^−1^ and can act as a suitable guest molecule for β-CyD (Figure 3IV) [97,98]. Benzimidazole can be protonated with one charge at acidic conditions. By protonation or deprotonation of benzimidazole upon varying pH, the host-guest interaction between CyD and benzimidazole can be mediated. At normal physiological pH of 7.4, benzimidazole can stably be included into the cavity of β-CyD because of its hydrophobic nature. However, when the benzimidazole is protonated in an endosomal/lysosomal atmosphere, it is hard to form host-guest inclusion with CyD further. Therefore, pH-responsive drug release systems can be constructed using the CyD-benzimidazole system. 

Another alkyl guest with pH responsive property is the dansyl group. The dansyl/β-CyD system is also based on the inclusion behavior of CyDs, which can capture the dansyl fluorescent group and further increase its fluorescence intensity. The basic imine group in the dansyl group can also be protonated under acidic conditions and deprotonated at higher pH values following a similar mechanism. At the same time, the dansyl group is accompanied with hydrophilic/hydrophobic variations and thus it could escape from the cavity of β-CyD.

Additionally, other stimulus-responsive switches like enzyme, thermal, and glucose could be integrated into host–guest inclusion systems. For example, H_2_O_2_ responsive fluorescein molecule FL2 is also used for a stimulus responsive switch [72]. Meanwhile, H_2_O_2_ changed the fluorescence and disassociated the inclusion of the host-guest cell with the release of the loaded drug captopril. Using semitransparent zebrafish as a model, non-invasive imaging of the beating heart reduced the heartbeat rate in vivo. The as-prepared NPs were effective in treating oxidative stress-induced heart failure, indicating oxidative stress may be a promising therapeutic intervention.

Thermal is also a kind of stimulus-responsive switch. At higher temperature, the total fraction of host-guest inclusion decreases because of weak non-covalent interactions involved in the inclusion behavior, such as hydrogen bonding, van der Waals forces, electrostatic interactions, dipole forces, and hydrophobic interactions. High temperature is expected to affect the stability of inclusion complexes and enhance their formation kinetics, resulting in smaller complex aggregates [99]. 

### 4.4. Self-Assembly Capability

This part will focus on self-assembly driven by inclusion complexation and CyD-related self-assembly. However, since dissociation takes place too rapidly upon dilution, ultimately release may take place during administration to the patient, so that self-assemble into NPs is a good strategy to realize sustained drug release [100]. Different interactions could co-exist in the formation of carriers when multicomponent systems containing CyD work as building blocks. Driven by interactions in specific regular, the unassociated and disordered components come together, resulting in various kinds of morphologies including cylinders, spheres, bicontinuous structures, lamellae, vesicles, and hierarchical assemblies. Supramolecular self-assembly is driven by non-covalent interactions, and affects the distribution of the components in systems [6]. The conjugated polymers or functional groups could graft to the reactive hydroxyl groups (primary or secondary) of the CyD mainly by chemical modification. Their ability to self-assemble is regulated by electrostatic forces, hydrogen bonds, van der Waals, and host–guest interactions. Generally, self-assembled CyD-based supramolecular systems derive various kinds of NPs such as micelles, uni/multilamellar emulsion bubble, nanospheres, nanosheets, nanogels, CyDplexes, etc. These have been verified to possess specific physicochemical and drug delivery features, especially as their small micro differences lead to macro differences. For example, size, charge, surface hydrophilicity, and the nature and density of the ligands on their surface impact the circulating half-life and in vivo fate of the nanoplatform [101]. 

CyDs could corporate with various linear, branched, cationic, anionic, copolymer, and co-block modular to form self-assembled nanoassemblies. Hydroxyl groups of CyDs that aligned on the surface of the truncated cone are important in grafting with other functional molecules. The CyDs were modified by several typical chemical reactions (such as amination, halogenation, esterification, and sulfonation) and introduced amine, halogen, alkoxy and sulfite groups that can be further modified or self-assembled [8,97,102]. Therefore, the intrinsic ability of CyD allows the synthesis of a series of self-assembled supramolecular structures with different functions via noncovalent interactions. 

#### 4.4.1. Self-Assembly Directed by Hydrophilic-Hydrophobic Interactions

Self-assembly based on hydrophilic-hydrophobic interactions is a common strategy to construct hydrogels, micelles, and NPs. Polymer vesicles or polymersomes consist of a “shell” and an inner structure in an aqueous solution, with the hydrophilic part on the outside and the hydrophobic core on the inside. Hydrophobic interactions are important nonspecific interactions in various systems that create hydrophilic-lipophilic structures and load hydrophobic oil, medicine, dye, and pollutions. The mechanism of hydrophobic interactions is related to the tentative redistribution of water molecules as hydrophobic parts tend to come close to each other. The hydrophilic–lipophilic balance is an important parameter in the construction of drug delivery systems. CyDs are amphiphilic compounds with water-soluble polysaccharide nature and relative hydrophobic inner cavities. Therefore, the as-formed host–guest inclusion complexes are also self-assemblies. From another point of view, amphiphilic CyDs, CyD-polymer, CyD- PRX, etc. have been synthesized with structure-property relationships, and some of their properties can be further modulated by various parameters, such as the number, type, and position of the as-used modules [6]. The hydrophobic modular, including the length of the hydrophobic chain and the connecting group (ester, ether, or amide), affects the interfacial properties of amphiphilic CyDs assemblies. The hydrophobic part could only aggregate under aqueous conditions. However, the amphiphilic part could interact with both the surrounding solution molecule and the hydrophobic part. A very extensively studied self-assembly process is the self-assembly of block or graft copolymers, where incompatible polymer chains are bonded covalently. 

Amphiphilic CyDs is one of the meaningful branches of modified CyDs and show unique characteristics for drug delivery. They effectively entrap both hydrophilic and hydrophobic drugs, solving their solubility, stability, and bioavailability problems [103]. Organized molecular structures can be built using amphiphilic derivatives, through self-assembling systems or by incorporation in lipid membranes, overcoming the drawbacks of nature CyDs as a result. The aim of synthesis of amphiphilic CyDs based on the following superiorities: enhance compatibility of CyDs with biological membranes, and form stable self-assemblies in aqueous solutions. In this case, the included drugs could protect from oxides or other unwanted interactions. One of the most important structural advantages of amphiphilic CyDs is that they can form nanosystems without addition of surfactants, avoiding the possible toxicity induced by these excipients [104]. 

Hydrophilic-hydrophobic self-assembly can be systematically acknowledged from elsewhere [104,105,106].

#### 4.4.2. Self-Assembly Directed by Charge Interactions

Charge interactions are associated with the spatial distribution of charges and play an important role in keeping the equilibrium of polyelectrolyte systems. The interaction energy is proportional to the number of charges carried by the oppositely charged groups. Charged monomers could determine the functions and properties of biological macromolecules such as DNA, RNA, proteins, and polysaccharides, which possess or could possess charge monomers [107,108]. Biocompatibility, inclusion complexation ability, and reduced systemic toxicity are all advantages of CyD grafting with ionic polymers. CyDs with positive charges can enhance antibacterial activity, while those with negative charges can improve antiviral activity and inhibit protein fibrosis [109]. From this point, this kind of material has profound compatibility with biomacromolecule systems and can serve as model systems for loading biological drugs (Table 2) [110,111]. Zhang et al. reported that positively charged amino groups from chitosan can prolong the corneal residence time and promote penetrability to aqueous humor based on the negatively charged cornea and conjunctiva [70]. The prepared NPs had no obvious side effects on the rabbit’s eyes and showed better capability for prolonging the residence time than that of the control naringenin suspension sample (192.5 ng/mL versus 52.8 ng/mL at 1.5 h). At the same time, the as-prepared NPs significantly increase naringenin bioavailability in the aqueous humor.

The layer number of the positive and negative charges based on self-assembly, also known as layer-by-layer self-assembly, can be as much as 50 layers or even more. However, it seems that this strategy cannot be applicable in drug delivery because of instability. Therefore, if stability problem can be overcome, layer-by-layer self-assembly can be considered in drug delivery. In order to increase separate efficiency of electron-hole pairs and improve H_2_O_2_ content, Niu et al. made full use of the charge interactions between β-CyD-HA and the triethanolamine-protected Bi_2_Se_3_ NPs and introduction of β-CyD-HA into NPs, and then adamantine modified hemin could incorporate into the system successfully (Figure 4I) [114]. 

On the other hand, cationic CyD carriers have been welcomed for polypeptide/protein delivery based on ionic interaction [110]. Wang et al. introduced a modular approach to realize the hierarchical self-assembly of discrete metal-organic cages (MOC) into supramolecular NPs. PEI could interact with protein and the NPs keep stable in the presence of protein, enabling the encapsulation of protein for intracellular protein delivery (Figure 4II,III) [119]. 

The sustained release pattern of anticancer drugs appears to be the key to reducing hepatotoxicity. The charge interactions based self-assembly also possesses sustained release property. For example, Lakkakula et al. synthesized hierarchical nanoflowers composed of cationic-β-CyD as polymeric core and alginate and chitosan “petals” (Cat-β-CyD/Alg-Chi nanoflowers) used for carriers based on ionic-gelation technique for oral delivery of 5-Fluorouracil (5-FU) [127]. Comparing the as-prepared nanoflowers to the inclusion complex alone, the nanoflowers released two times more slowly.

#### 4.4.3. Self-Assembly Directed by Coordination Interactions

Coordinate bonds, also known as donor-acceptor interactions, exist between electron donors and acceptors, such as π-donors/organometallics or Lewis bases/acids. This interaction is widespread in material systems and is widely for catalysis or assembly. The motivation to coordinately drive self-assembly to obtain functional materials stems from a set of inherent properties of the assemblies. The first advantage is that assembly dimensions can easily be changed without significantly altering the synthesis protocol. Second, metal-ligand bonds are generally more stable than those of other noncovalent interactions. Third, the control over position and number of noncovalent interaction moieties makes it possible to further explore the structure–property relationship and construct materials that we want. Moreover, the coordination interactions could integrate different types of components into the system. Finally, the internal cavities could also include a series of functional molecules, which further widen the application [23].

MOFs are one kind of coordination-driven self-assembly materials composed of metal ions or clusters linked by organic ligands, forming topological network structures. The metal-organic coordinated framework is composed of three parts, the metal ion/cluster, the p-block elements, and bridges [2]. The as-formed duplicate units demonstrate remarkable porosity which makes it suitable for a wide variety of applications, including in healthcare [107]. Compared with thermal- and light-stimulus responsibility, MOF-based drug carriers are more sensitive to acid stimulation for cancer and inflammation treatment due to their weakly acidic environment and fragile coordination bond. However, the conventional MOFs are made from the metal ion and toxic organic linker that are not safe for biological treatment in pharmaceutical applications. 

CyD MOFs have been proposed to decrease the potential health threats associated with the MOFs. Due to the presence of -OCCO- single bond binding groups in the primary and secondary faces, CyDs can be easily formed complexes with alkali and alkaline earth metal ions [128]. MOFs derived from γ-CyD show high specific surface area and have been used to prepare biocompatible and non-toxic MOFs. However, CyD MOFs are humid and unstable and dissolve in water. To overcome the decomposition or dissolution problems upon exposure to water and to further improve the drug loading efficiency, Jia et al. modified poly(ethylene glycol) dimethacrylate (PEGMA) via SI-ATRP onto the surface of γ-CyD-MOF (Figure 5I) [129]. As expected, the introduction of the PEGMA layer endowed pH responsive capability, enhanced water stability, and provided better biocompatibility. The modified γ-CyD-MOF carrier illustrated a high DOX loading efficiency of 89.1% with excellent targeting ability. Toxicity from metal ionic cannot be ignored in biomedical applications. To overcome this drawback, Xue et al. synthesized a new crosslinking molecule named dithiobis (propanoyl chloride) (DTPC) to functionalize the γ-CyD-MOF (Figure 5II) [130]. The contained disulfide bond possesses GSH responsive whereas the acyl chloride group accelerates the reaction with γ-CyD-MOF. After exposure to a humid atmosphere, the crosslinked CyD-MOFs were transformed into the cubic gel NPs (ssCGP) after lack of the potassium ion. The porous structure contributes to the large surface area and exhibited excellent drug-loading capability. Thus CyD-MOF-based materials with high specific surface area and superior safety are expected to be used as smart drug delivery vehicles.

#### 4.4.4. CyDs-Based Supramolecular Necklaces

CyDs-based supramolecular necklaces are polyCyD conjugates. In short, supramolecular formed through ring-like CyDs thread into a chain-like polymer via host-guest inclusion. PRX attaches a bulky group serving as a stopper at the end of chain axes so that the interlocked CyDs cannot slide out easily. Whereas polypseudorotaxane lacks a capping group at the two sides of the chain axes. Furthermore, CyDs that interlocked into circle polymer chains are named polycatenanes. The yield of rotaxane is low generally due to the lack of specific interactions between chain and ring-like structure molecules. Both the CyD rings and the polymer chains can be chemically modified to improve efficiency or cellular internalization. CyDs-based supramolecular necklaces have attracted considerable attention due to their unique topological structures and polyCyD nature. Due to their necklace structure, CyDs could both rotate and move along the axle chain freely while maintaining the necklace structure [131]. The formation of these CyD-based assemblies provides abundant binding sites in the resulting materials by further modifications. Hence the formed supramolecular necklaces could be further modified to endow many functions. 

The inclusion of CyDs with those hydrophilic guest polymer chains such as PEG results in a water-soluble host-guest inclusion supramolecular necklace [132,133]. By introducing hydrophobic drugs, the necklace could self-assemble by hydrophilic and hydrophobic interactions. Another interaction, such as hydrogen bonding between CyDs and hydrophobic drugs, strengthens the connection and results in increased drug loading content [131,134]. Apart from drug loading systems by host–guest inclusion of CyDs with guest molecules, a series of parameters such as viscosity, crystallization status, the encapsulated drug diffusion rate, the arrangement of the components, and water solubility of the formed supramolecular necklace could be altered [135,136,137]. The variation of the species of CyDs, the content of CyD(s), and the processing method could also vary the encapsulated content and release dynamics of the encapsulated drugs. That is to say, supramolecular necklace possesses great tenability and it may adapt to various biological environments.

The supramolecular necklaces could also behave like the aforementioned systems. However, there are also some differences. For example, gene transfer via nonviral vectors (transfection) is based on the incorporation of naked DNA but predominantly complexed with cationic polymers or cationic lipids (in polyplexes and lipoplexes) into the target population [138]. Due to this complexation, DNA cargo may be protected against degradation by nucleases and serum components by generating less negative surface charges, so that a plethora of pharmaceutical agents can be bound to the polymers to generate supramolecular prodrugs. Compared to the pre-mentioned CyDs-based polycations, the CyDs-based supramolecular necklaces are comprised of periodic and lamellar architectures. This suggests that compact supramolecular necklaces, such as nucleic acid nanostructures, can produce high transfection performance [108]. Besides, stability is another attractive point for CyDs-based supramolecular necklaces for DNA delivery. By tuning certain design features, such as cationic charge density, number of threaded CyDs, level of available free PEG moieties, size of PEG backbones, etc. (Figure 6I), Ji et al. have generated a series of multiarm PRX analogs [139]. This made it possible to establish nano quantitative structure–activity relationships to improve biodistribution, pharmacokinetics, and transfection efficiency. 

CyDs-based supramolecular necklaces shows a series of advantages such as good biocompatibility, abundant derivable hydroxyl groups, and tunable nanoscale size and chemical composition in drug delivery systems [140]. Furthermore, by adjusting the CyD position to fit external changes, supramolecular necklaces can effectively stabilize the system between supramolecular necklaces and cells because of the CyD rings can move freely on the polymer axle [34]. The abundant hydroxyl groups could be modified and endow various functions such as increase the solubility of the system, prolong the blood circulation time, decrease the cytotoxicity of the normal tissues, enhance the drug loading content, prevent drug leakage and increase the cytotoxicity of the tumor [131,141,142]. Liu et al. has reported that by the time CyD slides out from axle polymer, the formed CyD-deferoxamine conjugates (CyD-DFO) dissociated into constructs of approximately 2 nm for faster renal elimination (hydrodynamic diameter of less than 6 nm) (Figure 6II) [143]. Zhang et al. constructed a represented example that the grafting of D-a-Tocopheryl polyethylene glycol 1000 succinate (TPGS) and 10-hydroxycamptothecin (HCPT) onto α-CyD PRX [144]. The model anticancer drug, HCPT, was lytic and cytotoxic toward normal cells. The combination of TPGS and HCPT onto α-CyD PRX demonstrated that the as-prepared material is nontoxic to normal cells but effectively inhibits the growth of cancer cells.

**Figure 6 pharmaceutics-15-01536-f006:**
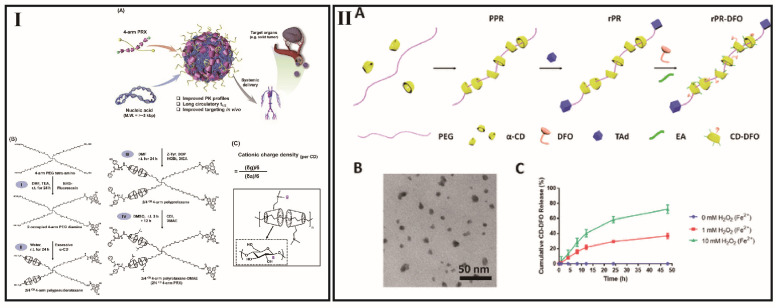
(**I**) (**A**) Schematic of four-arm PRX platform for systemic plasmid delivery; (**B**) The synthesis routes of four-arm PRX; (**C**) Optimizing the cationic charge density on α-CyD rings [139]; (**II**) (**A**) Schematic illustration for preparing PRX. (**B**) Representative TEM image of the prepared PRX. (**C**) Cumulative release at various H_2_O_2_ concentrations in the presence of 1 μM FeSO_4_ [143].

It is possible to include two axis polymer chains in one CyD cavity. High-molecular-weight linear PEGs can effectively interact with CyD to form pseudopolyrotaxanes and then gels [145]. Besides, CyD-based polypseudorotaxane hydrogel structures were also expanded by branched or grafted polymers. Supramolecular chemistry could also be engaged to modulate the rheological properties of its physical interactions [146]. One of its representative characteristics is reversibility. Its low rheological properties can be varied by the inclusion of CyD with guest PEG. This results in sol gel transition and the increase in rheological properties, which could be further applied to injectable hydrogel that plays an important role in tissue engineering [147,148,149]. Moreover, it has been reported that compared to commercial antibody-based drugs, the pseudopolyrotaxane drug-loaded hydrogel prepared by Higashi et al. demonstrates a stable advantage and also shows good safe profiles [149].

### 4.5. Fiber Formation

Intra/intermolecular interactions from polar groups in CyDs have proper chemical and physical properties. Supramolecular interactions have included many kinds of guest molecules in the cavity of CyDs and made apparent progress. The host and guest molecules could connect by supramolecular interactions and when the host and guest molecules were further connected by covalent bond, the system could connect continuously and theoretically form a fiber [150]. Apart from supramolecular interactions, the hydrogen bonding from intra/intermolecular could also affect the interactions originating from CyDs. Meanwhile, viscosity is an important factor affecting hydrogen bonding interactions. To increase the density of hydrogen bonding, ionic CyDs and highly soluble solvents are the two ideal options; a good example of its application is fiber formation. 

However, among the versatile fibers, electrospinning nanofiber is one of representative fibers in drug delivery. Celebioglu and coworkers prepared a series of fast-dissolving oral drug delivery nanofibrous webs and CyDs were used to improve the solubility of poorly water-soluble drugs (Figure 7) [151,152,153]. The presence of numerous hydrogen bonds between CyDs contributes to the formation of polymer-free CyD nanofibers [154]. Compared to traditional polymer electrospinning, in which chain entanglement makes the preparation easy, CyD rings form nanofibers by hydrogen bonds between the hydroxyl groups of the surrounding rings [155]. In these fast-dissolving systems, polymer-free electrospun nanofibrous mats can be spun successfully. The aim of polymer free strategy from these drug delivery systems is essential. This is because the fast-dissolving drug delivery system based on CyDs inclusion of complex nanofibers might be an ideal alternative to the electrospinning polymeric systems, as the complexing properties of CyDs can eliminate the potential toxicity of polymers [153,156]. Similarly, Topuz et al. also expressed worry about polymer-based electrospun nanofibrous mats possessing low loading capacity and the need to use toxic organic solvents to boost their antibiotic loading capacity [157]. Hence polymer free electrospun nanofibrous mats may be a potential biomaterial regarding their high drug-loading efficiency, green technological processes, and minimal side effects.

## 5. Summary and Outlook

In this review, we summarized the research progress of CyD-based nanoplatforms over the last five years and introduced their related effects in constructing biomaterials from the perspectives of structure, function, and application. Based on the structure of CyDs, scientists have widely explored the applications from their reactive hydroxyl group and their cavity for the inclusion of guest molecules. This has been further investigated and several applications have been derived, such as physicochemical characteristics alteration of the drugs, therapeutic promise, stimulus-responsive switch, self-assembly capability, and fiber formation. Despite the remarkable advances in the field, a great deal of effort is required to master the structure-property relationships and promote their practical applications. Conceptual and theoretical exploration is a prerequisite and it may be a basic support in the development of advanced CyD-based drug delivery materials, but also provides references for clinical treatment. After a general understanding of recent advances in CyD-based nanoplatforms, the following four points may be attractive to researchers for further exploration: Self-assembly has been widely applied to many fields, scientists have developed different kinds of materials and discovered the mechanism of self-assembly including charge interactions, hydrophilic-hydrophobic interactions, coordination interactions, etc. Benefiting from controllability, self-assembly should be further developed by scientists over a long time.Multifunction is another essential requirement for drug delivery. Regardless of the therapeutic effects of biomaterials, targeting capability, immune clearance avoidance, and biocompatibility are essential characteristics of the delivery system. The reactive hydroxyl group and cavity assigned to CyD are ideal candidates to meet the three needs at the same time. In-depth development of CyDs may help to increase the functions of drug delivery.Chemical modification, polypeptide modification, and biofilm functionalization are three powerful strategies in drug delivery. Among them, biofilm functionalization seems to be the most biocompatible one and has demonstrated great potential for clinical verification. However, biofilm functionalization is far from being explored in combination with CyDs.Artificial intelligence (AI) has influenced nearly every prospect of our human beings. The same could apply to the construction of CyD-based nanoplatforms. Insight sparked by AI in this field shortly [158].

In conclusion, the accumulated fascinating research works in the past 5 years have shown that CyDs can play versatile roles in the area of human health care. CyD functionalization could be valuable for increasing drug loading, improving the solubility, stability, permeability, absorption, bioavailability, and targeting capacity of drugs, and modifying drug release while retaining safety and efficacy. Indeed, more discoveries should be made to further promote the practical development of CyD-based drug delivery systems, which may allow us to fully understand the dynamic nature of biological events and bring about a positive and substantial influence on human health.

## Figures and Tables

**Figure 1 pharmaceutics-15-01536-f001:**
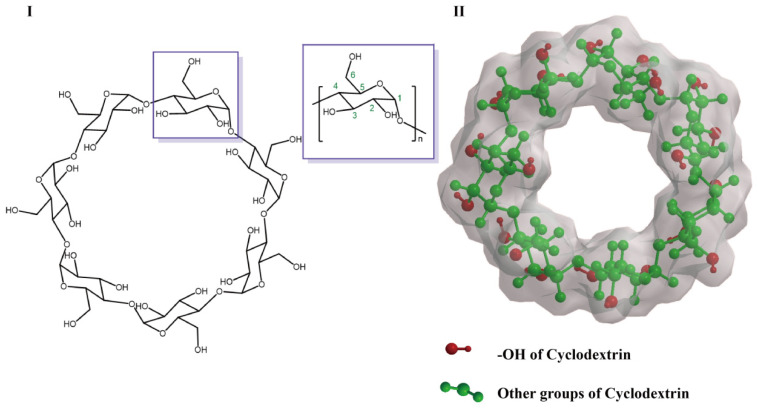
Representative structures of CyD (**I**,**II**). The surrounding gray area in (**II**) indicates the charge density derived from Chem 3D.

**Figure 2 pharmaceutics-15-01536-f002:**
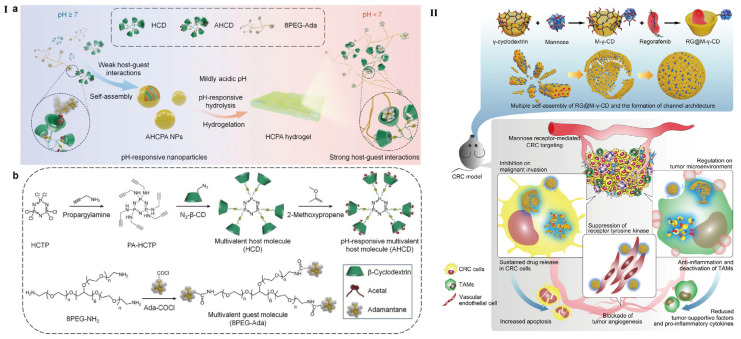
(**I**) (**a**) Schematic illustration of pH-responsive supramolecular NPs and hydrogel transmission after pH-responsive. (**b**) Schematic of synthesis route of multivalent host and guest molecules [44]; (**II**) Design of regorafenib@mannose-γ-CyD channel-type NPs (abbreviated as RG@M-γ-CyD CNPs) and the synergetic anti-colorectal cancer mechanism [45].

**Figure 3 pharmaceutics-15-01536-f003:**
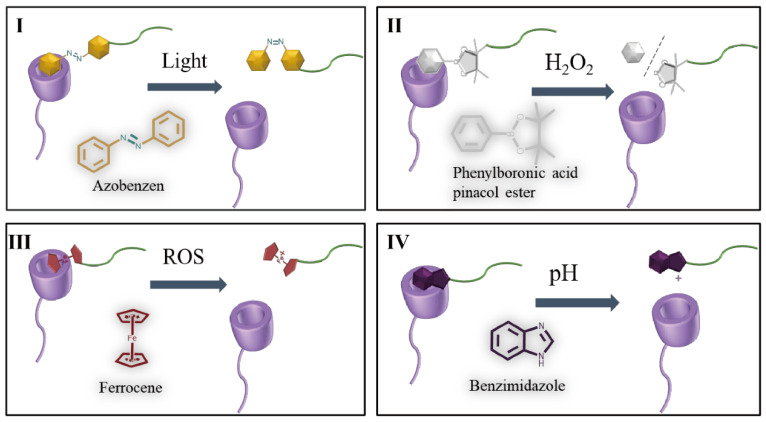
(**I**) Depiction of light responsive β-CyD-azobenzen complex; (**II**) Depiction of H_2_O_2_ responsive β-CyD-phenylboronic acid pinacol ester complex; (**III**) Depiction of ROS responsive β-CyD-ferrocene complex; (**IV**) Depiction of pH responsive β-CyD-benzimidazole complex.

**Figure 4 pharmaceutics-15-01536-f004:**
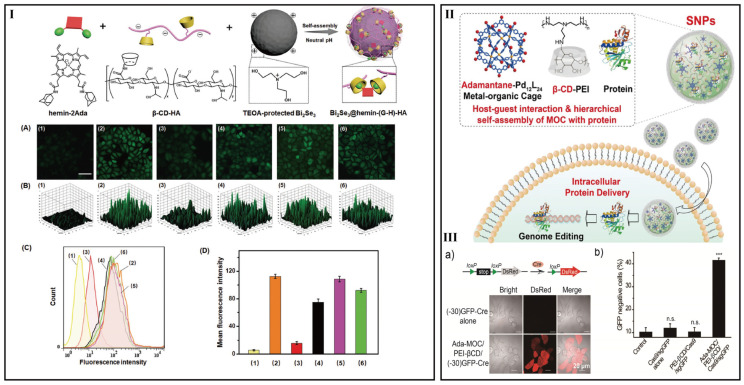
(**I**) Preparation of Bi_2_Se_3_@hemin-(G-H)-HA NPs for cooperative cancer therapy. (**A**) BES-H_2_O_2_ fluorescence images, (**B**) corresponding 2.5D images, (**C**) flow cytometric diagram, and (**D**) corresponding fluorescence intensity of HepG2 cells with different treatments: (1) Blank; (2) H_2_O_2_; (3) Bi_2_Se_3_@hemin-(G-H)-HA NPs + H_2_O_2_; (4) Bi_2_Se_3_ @hemin-HA NPs + H_2_O_2_; (5) Bi_2_Se_3_@HA NPs + H_2_O_2_; (6) hemin + H_2_O_2_ [114]. The scale bar is 100 µm. (**II**) Schematic of the self-assembly of adamantane-functionalized M_12_L_24_ MOC with β-CyD-conjugated polyethylenimine (PEI-βCyD) and interact with proteins into supramolecular NPs for intracellular protein delivery. (**III**) (**a**) Representation shows delivery of Cre deletes the stop cassette and activates downstream DsRed protein. (**b**) The green fluorescent protein expression profiles of the cells were quantified 48 h post protein delivery and compared to cells without treatment [119]. n.s. non-significant and *** *p* < 0.001 relative to the untreated control.

**Figure 5 pharmaceutics-15-01536-f005:**
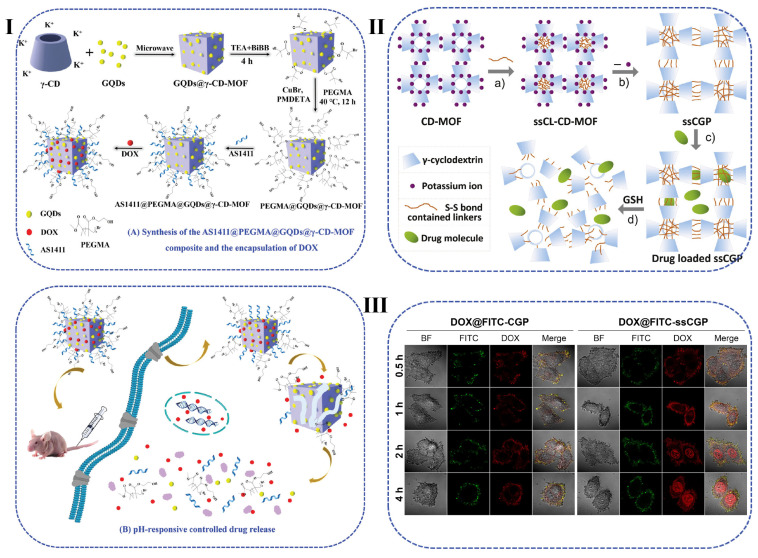
(**I**) Schematic illustration of the synthetic procedure for core-shell-structured PEGMA@GQDs@γ-CyD-MOF composite and PEGMA@GQDs@γ-CyD-MOF-based DOX loading and pH-responsive controlled release systems [129]. (**II**) Schematic illustration of the preparation of GSH responsive cubic gel particles. (**III**) CLSM of HepG2 cells treated with the as-prepared cubic gel after different incubation times (scale bar: 50 µm) [130].

**Figure 7 pharmaceutics-15-01536-f007:**
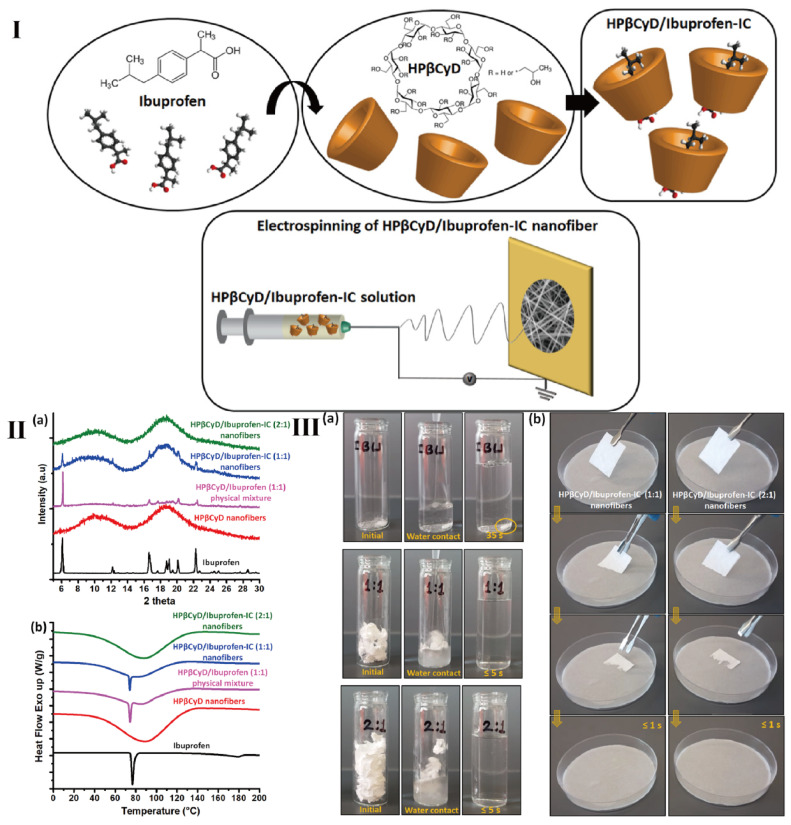
(**I**) Chemical structure of ibuprofen and HP-β-CyD, formation of inclusion complex between ibuprofen and HP-β-CyD molecules and the electrospinning of HP-β-CyD/ibuprofen-IC nanofibers. (**II**) (**a**) XRD and (**b**) DSC analysis of the as-prepared nanofibers. (**III**) (**a**) Dissolution behavior of as-prepared nanofibers. (**b**) Disintegration behavior of the as-prepared nanofibers at the artificial saliva environment [152].

**Table 1 pharmaceutics-15-01536-t001:** Stimulus responsive units based on CyDs in drug delivery.

CyD Type	Stimuli	Responsive Guest Molecules	Drug	Cells	Ref.
β-CyD	Light	Azo	DOX	MCF-7	[8]
β-CyD	NIR irradiation and reductase	Arylazopyrazole	siRNA	A549, HeLa, and 293T	[71]
β-CyD	light	Azo	Diclofenac sodium	MC3T3-E1	[72]
β-CyD	ROS	6-(Mercaptohexyl) ferrocene	β-Cyclodextrin/L-Arginine/Au Nanomotors	RAW264.7, and HUVECs	[73]
β-CyD	Light	Arylazopyrazole	Gold, iron oxide, and lanthanide-doped LiYF_4_ NPs		[74]
β-CyD	Light	Azo	Azo modified lanthanide upconversion NPs and β-CD modified downconversion nanoprobes	HEK293T and CaOV3	[75]
β-CyD	Hypoxia	Azo	Rho-TP	MCF-7	[76]
β-CyD	H_2_O_2_	Phenylboronic acid pinacol ester	DOX	4T1	[77]
β-CyD	ROS	Fc	CuS	B16	[78]
β-CyD	pH	Benzimidazole	DOX	MCF-7	[79]
β-CyD	ROS	Fc	DOX	4T1	[80]
β-CyD	pH	benzimidazole	β-CyD		[81]
β-CyD	H_2_O_2_	Fc	Glucose oxidase	CT26	[82]
β-CyD	Glucose	Phenylboronic acid	Insulin	L929	[83]
β-CyD	ROS	Fc	DOX	HepG2	[84]
β-CyD	ROS	Fc	DOX	HeLa	[85]
β-CyD	ROS	Fc	DOX	Bel-7402 and L02	[86]
β-CyD	ROS	Fc	CPT	HEK-293T and PC3	[87]
β-CyD	ROS	Fc	Platinum (IV)	4T1	[88]
β-CyD	ROS	Fc	Carboxy phthalocyanine	HT29 and A431	[89]

**Table 2 pharmaceutics-15-01536-t002:** Representative Self-Assembly Systems Based on Charge Interactions.

Strategy	Carrier	Drug	Target	Cells	Ref.
Dual stimulus responsive of NIR irradiation and reductase under anaerobic conditions	Upconversion NPs encapsulated by β-cyclodextrin-grafted hyaluronic acid/spermine modified with arylazopyrazoles-IC	siRNA	CD44	A549, HeLa, and 293T	[74]
layer-by-layer coating	Anionic-β-CyD and poly(acrylic acid) and poly(l-lysine)	Tetracycline		HGF and *S. aureus*	[107]
Host–guest inclusion forms the cationic supramolecular polymer	Cationic β-CyD/adamantane (Ad)-poly (vinyl alcohol) (PVA)-poly(ethylene glycol) (PEG) -IC	siRNA		A549 and A549/GFP	[112]
Improving Plasmid Transfection in 2D and 3D Spheroid Cells	Cationic hyper-branched cyclodextrin-based polymers	Plasmid DNA	EGFP	HT-29	[113]
Chemodynamic therapy photodynamic therapy	β-CyD-HA/Bi_2_Se_3_ NPs	Bi_2_Se_3_ and hemin-2Ada	NIR light assisted tumor targeting	HepG2	[114]
Compact polyelectrolyte complexes	β-CyD-functionalized chitosan/ alginate	Piroxicam	pH	RAW	[115]
Templated synthesis	CyD-nanoGUMBOS	IR780		MDA-MB-231, Hs578T, and MCF-7	[116]
Individually overcoming bio-barriers at each delivery stage	PEG-*b*-PLLDA/MSNs-SS-Py/CyD-PEI	DOX	P-glycoprotein	MCF7/ADR	[117]
Doubly linked aromatic clip–polycationic CyD hybrids	CyD-aromatic hybrid/plasmid DNA	Plasmid DNA		COS-7, and HepG2	[118]
Hierarchical self-assembly	β-CyD-conjugated polyethyleneimine with adamantane-functionalized M_12_L_24_ MOC-IC	Proteins		HeLa, HeLa-DsRed, and HEK-GFP	[119]
CyD conjugates with dendrimer	Glucuronylglucosyl-β-CyD conjugate	Cas9/single-guide RNA complex	Mouse brain	SHSY5Y	[120]
Protein co-assembly	Wind chime-like lysine-modified CyD	Ribonuclease A and deoxyribo- nuclease I	Nucleus	Hela	[121]
Layer-by-layer self-assembly	Cationic poly(CyD)/alginate	4-Hydroxy- tamoxifen	Pyrene and 4- hydroxy-tamoxifen	Immortalized mouse podocytes	[122]
Combinational therapy	Cationic poly (L-lysine) modified by β-CyD/PEGylated tetraphenyl- porphyrin (TPP)-IC	TPP-PEG and DNA		HeLa	[123]
Cationic moieties for targeted delivery and enhanced uptake	Cationic CyD magnetic nanocarrier	MTX	Magnetic	Saos-2 and human red blood cells	[124]
Intracellular protein delivery with fluorescent microscopy imaging	Tetraphenylethylene-featured metal-organic cages (MOCs) and β-CyD-conjugated polyethylenimine	Tetraphenyle- thylene and protein	MAPK/ERK signaling	Neural cells	[125]
Synergistic therapy	Multiple β-CyD-attached QD NPs/Ad-modified TCP1 peptide-targeting ligand	5-Fluorouracil and miRNA-34a mimics	Colorectal cancer	DLD1	[126]

## Data Availability

Not applicable.

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
