# Peer review of "The Role of Cyclodextrin in the Construction of Nanoplatforms: From Structure, Function and Application Perspectives"

_pharmaceutics, 2023, doi:10.3390/pharmaceutics15051536_

Round 1
Reviewer 1 Report
The author main abjective was to write a review on cyclodextrins. They mainly highlight stimulus-responsive CD and write about modern CDs produced by supramolecular methods.
A difficult task they have undertaken. The novelty value is not sufficiently emphasised. At the beginning of the review they write about very general information about CD. These are not necessary.The figures try to convey a lot of information, but they are not transparent, they are chaotic.
There are many typos and editorial errors. And a review of the literature used reveals a number of problems. The work of CD researchers without whom the review could not be published is not included. Ie. Szente-Szejtli, Lorenso, Helen-Parrot Lopez etc.
Careful rewriting is needed.
There are many typos and misspellings in the text.
Author Response
To reviewer #1:
The author main abjective was to write a review on cyclodextrins. They mainly highlight stimulus-responsive CD and write about modern CDs produced by supramolecular methods.
Comments: 1) A difficult task they have undertaken. The novelty value is not sufficiently emphasised.
Response: Thanks for the meaningful comments. We are sorry for our deficiency. We agree with this suggestion and made a series of improvements in the manuscript based on this comments. The novelty was re-writed in “Abstract”and “Introduction” sections. Below are details:
Abstract: Cyclodextrins (CyDs) in nano Drug drug delivery systems consist of cyclodextrins (CyDs) have kept constant attention for good compatibility, negligible toxicity, and improved pharmacokinetics of drugs. The unique hollow structureinternal cavity has endowed a lot of functions, such as inclusion of guest molecules, functional modification of active hy-droxyl groups, and noncovalent interactionswiden the application of CyDs in drug deliv-ery based on its advantage. Besides, the polyhydroxy structure has further extended the functions of CyDs by inter/intramolecular interactions and chemical modification. Fur-thermore, the versatile functions of the complex contribute to physicochemical characteris-tics alteration of the drugs, therapeutic talent, stimulus-responsive switch, self-assemble assembly capability, and fiber formation. This review attempts to list the recent progress interesting strategies of CyDs and discuss their roles in the drug delivery systemnanoplat-forms, which maybe a guideline for developing novel nanoplatforms. Future perspectives of on the construction of CyD-based drug delivery systemsnanoplatforms are also dis-cussed at the end of this review, which may be the possible directions direction for the con-struction of more rational and cost-effective delivery vehicles.
Part of Introduction: In the field of drug delivery, the following improvements should be accomplished: (i) improvement in treatment effect, (ii) minimum in drug leakage/toxicity in normal tissues or organs, and (iii) reduction in the intake dosage [11]. Several improvements can be achieved in a structural manner derived from the CyDs-based drug delivery system. In 2000-2004, FDA introduce α-CyD, β-CyD, and γ-CyD into the generally regarded as safe (GRAS) list for use as a food additive. CyDs are highly safe for animal and human ad-ministration as they have also been approved by the Food and Drug Administration (FDA). The first patent for CyDs in pharmaceutical formulations dates back to 1953 [12]. Over 70 CyDs-related programs are being studied for clinicians. At least six types of CyDs and over 89 pharmaceutical products based on CyD complexes have been for clinics [13, 14]. Oral, nasal, pulmonary, ocular, and parenteral routes have been used to deliver CyDs. α-CyD uptake for 13 weeks at 13.9 and 12.6 g per kg body weight per day did not cause any toxicity or adverse health effects [15]. Chemically modified CyDs have been synthe-sized to improve native CyDs' solubility, inclusion capability, controlled drug delivery capacity, and toxicity. The potentials of CyDs within drug delivery systems have moti-vated us to conduct in-depth exploration. However, there are still confusion about how to neatly utilize CyDs in the drug delivery system and how to use CyDs to achieve incredible therapeutic effects.
In this review, interesting directions and their potential in nanoplatforms based on CyDs have been discussed. We will also demonstrate engineered drug delivery systems designed through representative supramolecular, covalent interactions, and other inter-esting strategies related to CyD functions. Meaningful self-assembly, gene delivery, and stimulus responsive strategies have been listed and discussed. The current challenges in the field of drug delivery, breakthroughs in CyDs research, as well as future considera-tions and opportunities for the translation of CyDs-based materials into clinical practice were also highlighted.
Comments: 2) At the beginning of the review they write about very general information about CD. These are not necessary.
Response: Thanks for constructive comments. We are sorry for our deficiency. We have overviewed and re-examined the manuscript and found there are many innecessary information. Thus we deleted well-known information. The deleted contents are below:
The first paragraph of “2. structure”;
The 4, 5 and 6th paragraphs of “3.1 Noncovalent Interactions”;
The first paragraph of “4.1 Physicochemical Characteristics Alteration of the Drugs”;
Part of the first paragraph of “4.1.1 Without Any Modification”.
Details are in the manuscript.
Comments: 3) The figures try to convey a lot of information, but they are not transparent, they are chaotic.
Response: Thanks for meaningful comments! We are sorry for our deficiency. Based on this comment, we have simplified the Figures and the unimportant part has been deleted. In order to make the Figures intuitive and readable, some Figures have added frames. Below are details:
Figure 2 I) (a) Schematic illustration of pH-responsive supramolecular NPs and hydrogel transmission after pH-responsive. (b) Schematic of synthesis route of multivalent host and guest molecules [44]; II) Design of regorafenib@mannose-γ-CyD channel-type NPs (abbre-viated as RG@M-γ-CD CNPs) and the synergetic anti-colorectal cancer mechanism [45].
Figure 4 I) Preparation of Bi2Se3@hemin-(G-H)-HA NPs for cooperative cancer therapy. (A) BES-H2O2 fluorescence images, (B) corresponding 2.5D images, (C) flow cytometric diagram, and (D) corresponding fluorescence intensity of HepG2 cells with different treatments: (1) Blank; (2) H2O2; (3) Bi2Se3@hemin-(G-H)-HA NPs + H2O2; (4) Bi2Se3 @hemin-HA NPs + H2O2; (5) Bi2Se3@HA NPs + H2O2; (6) hemin + H2O2 [114]. II) Schematic of the self-assembly of adamantane-functionalized M12L24 MOC with β-CyD-conjugated polyethylenimine (PEI-βCyD) and interact with proteins into supramolecular NPs for intracellular protein delivery. III) (a) Representation shows delivery of Cre deletes the stop cassette and activates downstream DsRed protein. (b)  The green fluorescent protein expression profiles of the cells was quantified 48 h post protein delivery and compared to cells without treatment. The graph present mean ± SD. n = 3 repeats [119].
Figure 5 I) Schematic illustration of the synthetic procedure for core-shell-structured PEGMA@GQDs@γ-CyD-MOF composite and PEGMA@GQDs@γ-CyD-MOF-based DOX loading and pH-responsive controlled release system [129]. II) Schematic illustration of the preparation of GSH responsive cubic gel particles. III) CLSM of HepG2 cells treated with the as-prepared cubic gel after different incubation time [130].
Figure 6 â… ) (A) Schematic of 4-arm PRX platform for systemic plasmid delivery; (B) The synthesis routes of 4-arm PRX; (C) Optimizing the cationic charge density on α-CyD rings [139]; II) (A) Schematic illustration for preparing PRX. (B) Representative TEM image of the prepared PRX. (C) Cumulative release at various H2O2 concentrations in the presence of 1 μM FeSO4 [143].
Comments: 4) There are many typos and editorial errors.
Response: Thanks for the kind remind. We are very sorry for the negligence. We have checked the manuscript many times after modification. We believe there are no typos and editorial errors.
Comments: 5) And a review of the literature used reveals a number of problems. The work of CD researchers without whom the review could not be published is not included. Ie. Szente-Szejtli, Lorenso, Helen-Parrot Lopez etc.
Response: Thanks for the kind remind! We are very sorry for our negligence. We have read the works from the influencing authors and gained a lot of meaningful knowledge. These meaningful knowledge should also be realized by audiences. The cited works are listed below:
- Alvarez-Lorenzo, C.; Garcia-Gonzalez, C.A.; Concheiro, A.; Cyclodextrins as versatile building blocks for regenerative medicine. J. Control. Release, 2017, 268, 269-281.
- Redenti, E.; Szente, L.; Szejtli, J. Drug/cyclodextrin/hydroxy acid multicomponent systems. Properties and pharmaceutical applications. J. Pharm. Sci., 2000, 89, 1-8.
- Otero-Espinar, F. J.; Torres-Labandeira, J. J.; Alvarez-Lorenzo, C.; Blanco-Méndez, J.; Cyclodextrins in drug delivery sys-tems. J. Drug Deliv. Sci. Tec., 2010, 20, 289-301.
- Concheiro, A.; Alvarez-Lorenzo, C.; Chemically cross-linked and grafted cyclodextrin hydrogels: From nanostructures to drug-eluting medical devices. Adv. Drug Deliv. Rev., 2013, 65, 1188-1203.
- Hbaieb, S.; Kalfat, R.; Chevalier, Y.; Amdouni, N.; Parrot-Lopez, H.; Influence of the substitution of β-cyclodextrins by cationic groups on the complexation of organic anions. Mater. Sci. Eng. C, 2008, 28, 697-704.
- Perret, F.; Marminon, C.; Zeinyeh, W.; Nebois, P.; Bollacke, A.; Jose, J.; Parrot-Lopez, H.; Le Borgne, M.; Preparation and characterization of CK2 inhibitor-loaded cyclodextrin nanoparticles for drug delivery. Int. J. Pharm., 2013, 441, 491-498.
- Simoes, S.; Rey-Rico, A.; Concheiro, A.; Alvarez-Lorenzo, C.; Supramolecular cyclodextrin- based drug nanocarriers. Chem. Commun., 2015, 51, 6275-6289.
- Perret, F.; Duffour, M.; Chevalier, Y.; Parrot-Lopez, H.; Design, synthesis, and in vitro evaluation of new amphiphilic cy-clodextrin-based nanoparticles for the incorporation and controlled release of acyclovir. Eur. J. Pharm. Biopharm., 2013, 83, 25-32.
- Simões, S. M. N.; Rey-Rico, A.; Concheiro, A.; Alvarez-Lorenzo, C.; Supramolecular cyclodextrin-based drug nanocarriers. Chem. Comm., 2015, 51, 6275-6289.
- Taveira, S.; Varela-Garcia, A.; Dos Santos Souza, B.; Marreto, R.; Martin-Pastor, M.; Concheiro, A.; Alvarez-Lorenzo, C.; Cyclodextrin-based poly(pseudo)rotaxanes for transdermal delivery of carvedilol. Carbohyd. Polym., 2018, 200, 278-288.
- Marreto, R.; Cardoso, G.; Dos Santos Souza, B.; Martin-Pastor, M.; Cunha-Filho, M.; Taveira, S.; Concheiro A.; Alva-rez-Lorenzo, C.; Hot melt-extrusion improves the properties of cyclodextrin-based poly(pseudo)rotaxanes for transdermal formulation. Int. J. Pharm., 2020, 586, 119510.
- Di Donato, C.; Iacovino, R.; Isernia, C.; Malgieri, G.; Varela-Garcia, A.; Concheiro, A.; Alvarez-Lorenzo, C.; Polypseu-dorotaxanes of pluronic (R) F127 with combinations of alpha- and beta-cyclodextrins for topical formulation of acyclovir. Nanomaterials, 2020, 10, 613-628.
Comments: 6) Careful rewriting is needed.
Response: Thanks for the kind remind! We are very sorry for our deficiency. We have re-conducted the manuscript on a more strict standard to treat our manuscript. Correspongding variations have been marked in the manuscript.
Comments: 7) There are many typos and misspellings in the text.
Response: Thanks for your time for improving the quality of our manuscript! We are very sorry for our deficiency. We have checked the whole manuscript several times and we believe typos and misspellings disappeared. Details are in the manuscript.

Reviewer 2 Report
Dear Editor, Pharmaceutics
The manuscript seems well written and well presented. Figures and tables provided are appropriate and sufficient to explore the study. I have some minor corrections.
1. Please modify the title of the table II. It may be written as 'Representative self-assembly systems based on charge interactions'
2. Figure 5II, may be inserted in a box as like as 5I.
3. In section 5, Summary and Outlook, numbering (1-4) may be changed as separate points.
Manuscript may be considered for publication after minor corrections.
Minor corrections
Author Response
To reviewer #2:
Comments: 1) Please modify the title of the table II. It may be written as 'Representative self-assembly systems based on charge interactions'
Response: Thanks for the practical comments! We are very sorry for our deficiency. We have corrected the title of table II as “Representative self-assembly systems based on charge interactions”.
Comments: 2) Figure 5II, may be inserted in a box as like as 5I.
Response: Thanks for the meaningful comments! We are very sorry for our negligence. We have re-conducted the Figure 5 and all images are in the same formulation.
Figure 5 I) Schematic illustration of the synthetic procedure for core-shell-structured PEGMA@GQDs@γ-CyD-MOF composite and PEGMA@GQDs@γ-CyD-MOF-based DOX loading and pH-responsive controlled release system [131]. II) Schematic illustration of the preparation of GSH responsive cubic gel particles. III) CLSM of HepG2 cells treated with the as-prepared cubic gel after different incubation time [132].
Comments: 3) In section 5, Summary and Outlook, numbering (1-4) may be changed as separate points.
Response: Thanks for the meaningful comments! We are sorry for our negligence. We have changed the numbering in Summary and Outlook into separate points.

Reviewer 3 Report
This is a well-conducted research on the cyclodextrin. Very complete review paper with a scientifically sound distribution of the exposed material. The manuscript is written in a scholarship style, avoiding verbosity, that facilitates reading. in the literature there are many reviews on cyclodextrins, this one from a different view. I strongly support publication in Pharmaceutics after few minors, merely formal corrections:
1. In general, the introduction is too short. It seems right to me to suggest that you to amplify this part. On the basis of lines 47-49, the FDA has approved cyclodextrins but only for some routes of administration, specifying non-toxic doses. For this reason, I recommend quoting the EMA document (Cyclodextrins used as excipients. EMA/CHMP/333892/2013, 9 October 2017) and being more precise.
2. The title of your manuscript is based on “The role of cyclodxtrin in the construction of nanoplatforms”. For this reason, paragraph 4.4 needs to be deepened. Can you add these citation:
- De Gaetano, F et al. Amphiphilic Cyclodextrin Nanoparticles as Delivery System for Idebenone: A Preformulation Study. Molecules. 2023, 28, 3023; doi:10.3390/molecules28073023”
- Bonnet, V et al. Cyclodextrin Nanoassemblies: A Promising Tool for Drug Delivery. Drug Discov. Today 2015, 20, 1120–1126. doi: 10.1016/j.drudis.2015.05.008
- Musumeci, T et al. A Physico-Chemical Study on Amphiphilic Cyclodextrin/Liposomes Nanoassemblies with Drug Carrier Potential. J. Liposome Res. 2020, 30, 407–416. doi: 10.1080/08982104.2019.1682603
Describe in detail the ability of amphiphilic cyclodextrins to form supramolecular nanoassemblies in water. Specify that the presence of cyclodextrin nanoassemblies allows the administration of drugs insoluble in the physiological environment and increase their antioxidant activity, because it is an important goal.
After these necessary insights, for me the manuscript can be published.
Author Response
To reviewer #3:
Comments: 1) In general, the introduction is too short. It seems right to me to suggest that you to amplify this part. On the basis of lines 47-49, the FDA has approved cyclodextrins but only for some routes of administration, specifying non-toxic doses. For this reason, I recommend quoting the EMA document (Cyclodextrins used as excipients. EMA/CHMP/333892/2013, 9 October 2017) and being more precise.
Response: Thanks for the professional comments! We are sorry for our deficiency. The introduction part has been amplify. Especially the lines 47-49, they have been discussed in depth according to EMA document. Below are details:
Although traditional drugs have good therapeutic talent, they also face a series of drawbacks such as poor water-solubility, instability, short circulation time, unspecific targeting, and low biocompatibility, and thus they are difficult to be effective as we ex-pected in clinics [1, 2]. Hydrophobic drug molecules can accumulate in fat tissues, im-pairing patient recovery [3]. Additionally, 70% of new drug candidates are hydrophobic, which requires great efforts to formulate these water-insoluble drugs [4]. Drug administration in conventional dosage forms can result in unwanted therapeutic outcomes, such as drug toxicity/drug inefficiency and uncontrolled drug delivery. High drug loading and sustained/controlled release into targeted sites have led to the development of advanced drug delivery systems. Various strategies are used to improve pharmaceutical or bio-pharmaceutical delivery systems, such as interaction principle, assembly technologies, and/or targeted strategies [5]. Researchers have also developed many kinds of nanoplat-forms (such as polymer, lipid, amino acid, polypeptide, and inorganic-based platforms) for drug delivery and dramatic advances have been driven in chemistry, materials science, clinical practice and biotechnology [6, 7]. Many of the drug delivery systems have focused on controlled release therapeutics at appropriate times. They can be endowed to target specific locations within the body by active or passive targeting, thereby reducing the number of drugs to achieve equal therapeutic effect along with alleviated side effects to the patient [8-10]. The drug delivery systems also allow for more specific drug targeting and administration by tailoring the physicochemical and pharmacokinet-ic/pharmacodynamics properties of the original drugs to maximize therapeutic benefits.
In the field of drug delivery, the following improvements should be accomplished: (i) improvement in treatment effect, (ii) minimum in drug leakage/toxicity in normal tissues or organs, and (iii) reduction in the intake dosage [11]. Several improvements can be achieved in a structural manner derived from the CyDs-based drug delivery system. In 2000-2004, FDA introduce α-CyD, β-CyD, and γ-CyD into the generally regarded as safe (GRAS) list for use as a food additive. CyDs are highly safe for animal and human ad-ministration as they have also been approved by the Food and Drug Administration (FDA). The first patent for CyDs in pharmaceutical formulations dates back to 1953 [12]. Over 70 CyDs-related programs are being studied for clinicians. At least six types of CyDs and over 89 pharmaceutical products based on CyD complexes have been for clinics [13, 14]. Oral, nasal, pulmonary, ocular, and parenteral routes have been used to deliver CyDs. α-CyD uptake for 13 weeks at 13.9 and 12.6 g per kg body weight per day did not cause any toxicity or adverse health effects [15]. Chemically modified CyDs have been synthe-sized to improve native CyDs' solubility, inclusion capability, controlled drug delivery capacity, and toxicity. The potentials of CyDs within drug delivery systems have moti-vated us to conduct in-depth exploration. However, there are still confusion about how to neatly utilize CyDs in the drug delivery system and how to use CyDs to achieve incredible therapeutic effects.
In this review, interesting directions and their potential in nanoplatforms based on CyDs have been discussed. We will also demonstrate engineered drug delivery systems designed through representative supramolecular, covalent interactions, and other inter-esting strategies related to CyD functions. Meaningful self-assembly, gene delivery, and stimulus responsive strategies have been listed and discussed. The current challenges in the field of drug delivery, breakthroughs in CyDs research, as well as future considera-tions and opportunities for the translation of CyDs-based materials into clinical practice were also highlighted.
Comments: 2) The title of your manuscript is based on “The role of cyclodxtrin in the construction of nanoplatforms”. For this reason, paragraph 4.4 needs to be deepened. Can you add these citation:
-De Gaetano, F et al. Amphiphilic Cyclodextrin Nanoparticles as Delivery System for Idebenone: A Preformulation Study. Molecules. 2023, 28, 3023; doi:10.3390/molecules28073023”
-Bonnet, V et al. Cyclodextrin Nanoassemblies: A Promising Tool for Drug Delivery. Drug Discov. Today 2015, 20, 1120–1126. doi: 10.1016/j.drudis.2015.05.008
-Musumeci, T et al. A Physico-Chemical Study on Amphiphilic Cyclodextrin/Liposomes Nanoassemblies with Drug Carrier Potential. J. Liposome Res. 2020, 30, 407–416. doi: 10.1080/08982104.2019.1682603.
Response: Thank you for your helpful comments! We appreciatethe time you paid to the manuscript. The paragraph 4. 4 has been discussed in detail. Details are in the manuscript:
Meanwhile, the references have been carefully investigated and cited in the manuscript. Details are below:
- De Gaetano, F.; Scala, A.; Celesti, C.; Lambertsen Larsen, K.; Genovese, F.; Bongiorno, C.; Leggio, L.; Iraci, N.; Iraci, N.; Mazzaglia, A.; Ventura, C. A.; Amphiphilic cyclodextrin nanoparticles as delivery system for idebenone: a preformulation study. Molecules, 2023, 28, 3023.
- 102. Bonnet, V.; Gervaise, C.; Djedaini-Pilard, F.; Furlan, A.; Sarazin, C.; Cyclodextrin nanoassemblies: a promising tool for drug delivery. Drug Discov. Today, 2015, 20, 1120-1126.
- 104. Musumeci, T.; Bonaccorso, A.; De Gaetano, F.; Larsen, K. L.; Pignatello, R.; Mazzaglia, A.; Ventura, C. A.; A physi-co-chemical study on amphiphilic cyclodextrin/liposomes nanoassemblies with drug carrier potential. J. Liposome Res., 2019, 30, 407-416.
Comments: 3) Describe in detail the ability of amphiphilic cyclodextrins to form supramolecular nanoassemblies in water. Specify that the presence of cyclodextrin nanoassemblies allows the administration of drugs insoluble in the physiological environment and increase their antioxidant activity, because it is an important goal.
Response: Thanks for constructive comments! We are very sorry for our deficiency. Amphiphilic cyclodextrins is a very important part of self-assembly. We added the discussion about amphiphilic cyclodextrins in the manuscript. Details are below:
Amphiphilic CyDs is one of the meaningful branch of modified CyDs and show unique characteristics for drug delivery. They effectively entrap both hydrophilic and hydrophobic drugs, solving their solubility, stability, and bioavailability problems [103]. Organized molecular structures can be build using amphiphilic derivatives, through self-assembling systems or by incorporation in lipid membranes, overcoming the drawbacks of nature CyDs as a result. The aim of synthesis of amphiphilic CyDs based on the following superiorities: enhance compatibility of CyDs with biological membranes, and form stable self-assemblies in aqueous solutions. In this case, the included drugs could protect from oxides or other unwanted interactions. One of the most important structural adantage of amphiphilic CyDs is that they can form nanosystems without addition of surfactants, avoiding the possible toxicity induced by these excipients [104].

Round 2
Reviewer 1 Report
The manuscript is quite better than the pervious one.
Minor editing of English might be made.